# Network analysis of the human structural connectome including the brainstem

**Salma Salhi**[1,2]☯*, **Youssef Kora**[1,2,4]☯, **Gisu Ham**[1,2,3], **Hadi Zadeh Haghighi**[1,2,4], **Christoph Simon**[1,2,4]

**1** Department of Physics and Astronomy, University of Calgary, Calgary, Alberta, Canada, **2** Hotchkiss Brain Institute, University of Calgary, Calgary, Alberta, Canada, **3** Fishtank Consulting, Calgary, Alberta, Canada, **4** Institute for Quantum Science and Technology, University of Calgary, Calgary, Alberta, Canada

☯ These authors contributed equally to this work.
\* salma.salhi@ucalgary.ca

**Data Availability Statement:** The data cannot be shared in our submission since users of Human Connectome Project (HCP) data are not allowed to share it, as per the data use agreement. However, the data are publicly available from the HCP

## Abstract

The underlying anatomical structure is fundamental to the study of brain networks, but the role of brainstem from a structural perspective is not very well understood. We conduct a computational and graph-theoretical study of the human structural connectome incorporating a variety of subcortical structures including the brainstem. Our computational scheme involves the use of Python DIPY and Nibabel libraries to develop structural connectomes using 100 healthy adult subjects. We then compute degree, eigenvector, and betweenness centralities to identify several highly connected structures and find that the brainstem ranks highest across all examined metrics, a result that holds even when the connectivity matrix is normalized by volume. We also investigated some global topological features in the connectomes, such as the balance of integration and segregation, and found that the domination of the brainstem generally causes networks to become less integrated and segregated. Our results highlight the importance of including the brainstem in structural network analyses.

## Introduction

Network-based approaches are exceedingly useful tools for investigating the computational and informational power of the brain [1]. The core idea involves the reduction of an object as complex and generally intractable as the brain to a much simpler system of basic interacting neuronal elements [2]. This profound simplification has proven to be quite powerful, leading to insights into a variety of neuroscientific topics and applications [3]. For instance, diseases such as Alzheimer's, schizophrenia, and Parkinson's disease can be seen as disorders of brain networks [4]. The emerging field of network neuroscience [5] is working towards a theoretical framework which unifies the principles underlying the wide range of observed neurobiological phenomena, and deepens our understanding of the relationship between brain structure and function [6].

The goals of such network-based methods include the mapping, modeling and analysis of brain networks, both at a structural and functional level. Structural brain networks are constructed to reflect the direct fibre connections between the different anatomical regions, while

database, which is available at this URL: https://db.humanconnectome.org/app/template/Login.vm.

**Funding:** All authors of our paper (Salma Salhi, Youssef Kora, Gisu Ham, Christoph Simon, and Hadi Zadeh Haghighi) were funded by the NSERC Discovery Grant RGPIN-2020-03945, URL: https://www.nserc-crsng.gc.ca/index_eng.asp. The funder (NSERC) had no role in study design, data collection and analysis, decision to publish, or preparation of the manuscript.

**Competing interests:** The authors have declared that no competing interests exist.

functional networks refer to relationships between components which are statistically correlated regardless of the strength of their anatomical connections [7]. Studies of the latter are quite ubiquitous in the literature. These networks are typically extracted through techniques such as electroencephalography (EEG) and functional magnetic resonance imaging (fMRI), which enable the investigation of the topological properties of the dynamical patterns that emerge, both in the resting state and during the performance of tasks [8–10]. The study of structural networks, on the other hand, has gained prominence in more recent years, largely owing to the recent advances in non-invasive techniques that allow the imaging and mapping of the structural connectome [11, 12].

The human structural connectome provides a foundation for neurobiological research [13]. Indeed, the structural wiring of the brain constrains the cost of controlling a particular brain state; the transition to higher information content states is more costly and is modulated by the anatomical structure [14]. Functional networks are influenced by their structural counterparts; it has been shown that the strength and persistence of functional connectivity is constrained by the anatomical cortex [15], amongst other key factors such as local dynamics [16] and neuromodulation [17]. In fact, it is thought that anatomical connectivity allows for the reconciliation of the opposing requirements for functional networks, integration and segregation [18], where integration refers to the ability to quickly combine information from distant brain regions, and segregation refers to the ability for specialized processing to occur within dense regions [19].

It has been proposed that the brainstem is necessary for "core consciousness"—the simplest form of consciousness and the self—because the somato-sensory nuclei in the brainstem are best suited for its modulation [20]. Some theorists maintain that the regulation and arousal of consciousness are affected by the same part of the brain, which is thought to be the brainstem [21], and some even suggest that the brainstem alone can keep a subject conscious [22]. As we are also interested in the applications of structural connectomics to theories of consciousness, this provided motivation for the consideration of the brainstem in our structural network analysis.

We designed a computational fibre tractography method in Python using DIPY and Nibabel libraries to construct the structural connectomes. We constructed two connectomes: one containing 104 structures that includes the brainstem and is based on the parcellated images provided by the Human Connectome Project (HCP), and another comprising only 84 structures, which excludes some important subcortical regions including the brainstem and is comprised of the structures typically found in the MRTrix Desikan-Killiany atlas, which is the default atlas used in MRTrix, the popular software for brain network studies. We shall henceforth refer to the former as the "extended connectome", and the latter as the "restricted connectome".

With the networks obtained, we identified the most structurally important brain regions as those corresponding to nodes occupying the most central positions in each network, i.e., hubs. There are a number of graph-theoretical measures to characterize the centrality of a given node [23]. We started by the simplest of all, degree centrality (also known as node degree, node weight, or connectivity strength), which is defined as the sum of all connections to the node in question. In the case of a weighted network such as ours, this would simply correspond to the total number of streamline connections associated with each brain region. Such a quantity measures the extent to which a given node has strong connections to other nodes in the network, irrespective of their importance.

The importance of other nodes is taken into account by computing the eigenvector centrality [24, 25], which has been applied in more recent times to fMRI data in the human brain [26, 27]. This measure privileges nodes that are well-connected to other nodes in the network

which are themselves strongly connected. As the name suggests, this is accomplished by the eigenvector decomposition of the connectivity matrix.

The third measure we computed for our networks is betweenness centrality [28]. Unlike the aforementioned types of centrality, which directly quantify the connectivity of a given node to the rest of the network, betweenness centrality measures the extent to which the node is strategically located in the network. This is achieved by considering the number of shortest geodesic paths between all possible pairs of nodes that pass through the node of interest. Its computation is of complexity $O(n^3)$, which renders it an unfeasible measure for sufficiently large networks. It has, however, been used in the context of brain networks with a moderate number of nodes [29, 30], and since the sizes of the networks considered here do not exceed 104 regions, we did not contend with any serious computational limitations.

We also studied some global topological features in the different types of networks, such as the balance of integration and segregation. Integration was estimated by computing the global efficiency, which quantifies the extent to which a graph is interconnected by calculating the average over all pairs of vertices of the inverse of the shortest distance between them [3]. Segregation was quantified by relying upon the notion of modularity, which measures the extent to which a network may be partitioned into modules that are densely connected as compared to a null model of random connections [31]. Finally, we compute the global reaching centrality, which is a measure of the amount of hierarchical architecture within a graph.

Several results came to light after applying these graph-theoretical tools to the restricted and extended connectomes. Most notably, we found that the brainstem scores highest across all centrality measures, followed by the superiorfrontal cortices and the right and left thalami. The removal of the 20 subcortical structures in the restricted connectome, which is modelled from a well-known atlas, caused the ranking of the prominent structures to change *on average*, albeit with substantially overlapping distributions. Additionally, the exceedingly dominant presence of the brainstem seems to generally cause networks to become both less integrated and less segregated. We also observed that, following a normalization procedure intended to take into account the substantial volume of certain structures, the brainstem retained, across all centrality measures, quite a high place among the top structures notwithstanding its great size.

## Materials and methods

### Data analysis

No new data was collected for this project. Data was obtained from the Human Connectome Project (HCP) database. The HCP project (Principal Investigators: Bruce Rosen, M.D., Ph.D., Martinos Center at Massachusetts General Hospital; Arthur W. Toga, Ph.D., University of Southern California, Van J. Weeden, MD, Martinos Center at Massachusetts General Hospital) is supported by the National Institute of Dental and Craniofacial Research (NIDCR), the National Institute of Mental Health (NIMH) and the National Institute of Neurological Disorders and Stroke (NINDS). HCP is the result of efforts of co-investigators from the University of Southern California, Martinos Center for Biomedical Imaging at Massachusetts General Hospital (MGH), Washington University, and the University of Minnesota. We used data from 100 adult subjects ranging in age from 22 to 35 years, with 44 females and 56 males, all from the "WU-Minn HCP Data—1200 Subjects" round of data collection. Since the data is publicly available, we did not need to seek approval by an ethics committee or institutional review board for the collection or use of this data. According to HCP protocol, all data is obtained with written informed consent and the data is anonymized. Both preprocessed T1-weighted structural images and 3T dMRI images were used in our computational fibre

tracking method. The data was preprocessed by the HCP according to their standard preprocessing protocol. Python DIPY and NiBabel libraries were utilized to perform the streamline calculations using a constrained spherical deconvolution model and probabilistic fibre tracking functions, which are built in the libraries.

MRTrix is perhaps the most widely used software in brain network studies, but the atlas most readily available to its users is the FreeSurfer Desikan-Killiany atlas, which consists of only 84 structures and does not include the brainstem. The full list of these structures can be found in S1 Table. This atlas has been used in studies of structural-functional relationships (see, for instance, [32–34]). The brainstem is conspicuously missing in this atlas, so we designed a computational fibre tractography method in Python using DIPY and Nibabel libraries to calculate a connectivity matrix that would include the brainstem. Using these algorithms, we were able to extract 104 structures from dMRI images provided by the Human Connectome Project (HCP). We constructed two connectomes: one containing the 104 structures extracted using our Python method, and the other comprising the 84 structures typically found in the default MRTrix Desikan-Killiany atlas.

A white matter mask was first created by extracting labels of structures provided in the HCP structural data. Regions that matched the labels for white matter were identified, and the white matter mask was then used to calculate the seeds from which to begin the fibre tracking. For the extended connectome, the mask included the brainstem and a few other structures commonly omitted, the full list of which is found in Supporting information. A response function, which is used to form the constrained spherical deconvolution (CSD) model, was then calculated using the DIPY `auto_response_ssst` function. The CSD model was then combined with the structural white matter mask to generate a generalized fractional anistropy (GFA) model, which would aid in the fibre tracking process. A probabilistic direction getter model was then created using a maximum angle of 45˚, and this was finally used along with seeds and the diffusion affine matrix to generate a list of streamlines, using the DIPY `LocalTracking` class. Note that this function does not contain a parameter specifying the maximum number of streamlines, but instead only has a parameter specifying the maximum number of steps taken to generate streamlines (to prevent infinite loops) which is 500 steps by default. The function also uses the direction getter, the affine matrix, the seeds, and the stopping criterion to generate the streamlines. For each subject, the total number of streamlines is thus on the order of $10^4$. A list of endpoints was generated by extracting the first and last elements in each streamline list. This was then used in conjunction with the label mask to assess which structures each endpoint corresponded to in order to create a bi-directional label map. This is essentially a connectivity matrix that matches the total number of streamlines associated with a structure to the correct endpoint.

Due to the probabilistic nature of this tracking algorithm, the program was run three times for each subject and the results were averaged across all three runs in order to corroborate the results of the runs and ensure accuracy. These averages were then collected and averaged across 100 subjects to produce one overarching averaged connectome, representative of all subjects. All python code files are provided on the Github repository created for this project, which can be found at this link: https://github.com/SalmaSalhi7/Structural-Connectome-Project.

The same method was used to calculate the connectivity matrix with a reduced number of structures. The white matter masks were altered to eliminate undesired structures, enabling the fibre tracking algorithm to only examine the 84 structures in the restricted connectome, the full list of which can be found in S1 Table. A list of the 20 eliminated structures can be found in S2 Table.

## Centrality measures

We computed degree centrality simply as the sum of all connections to a given node

$$d_i = \sum_{i=1}^{N} a_{ij} \tag{1}$$

where $a_{ij}$ are the elements of the structural connectivity matrix $A$. The next examined metric is eigenvector centrality, defined as the components of the eigenvector $\mathbf{v}$ corresponding to the largest eigenvalue of $A$. The signficance of such a quantity may be seen by looking at the eigenvalue equation

$$\mathbf{v} = \frac{1}{\lambda} A\mathbf{v} \tag{2}$$

where $\lambda$ is the largest eigenvalue of $A$. By expanding the matrix multiplication process, one may write each component of $\mathbf{v}$ as

$$v_i = \frac{1}{\lambda} \sum_{i=1}^{N} a_{ij} v_j \tag{3}$$

which is similar to Eq 1, but the sum is now weighted by the scores of the node. These scores are all guaranteed to be positive by the Perron-Frobenius theorem, provided that $A$ is non-negative and irreducible, i.e., it has at least one non-zero off-diagonal element in each row and column (which will always be true in the cases of interest). These values are only unique up to an overall multiplication factor which depends on the choice of normalization [35], but such a factor does not play a role in the process of comparing nodes within the same network, which is our goal in this study. Finally, betweenness centrality was defined as

$$b_i = \frac{2}{(N-1)(N-2)} \sum_{i \neq j, i \neq k, j \neq k} \frac{\sigma_{jk}(i)}{\sigma_{jk}} \tag{4}$$

where $\sigma_{jk}$ is the number of shortest paths between nodes $j$ and $k$, and $\sigma_{jk}(i)$ is the number of shortest paths between nodes $j$ and $k$ that pass through node $i$. To define the shortest paths in a complete weighted graph such as ours, a notion of distance is required, which was simply taken to be the reciprocal of the weight. We employed the Python package NetworkX [36] in our computations of the betweenness centrality.

Together, $d_i$, $v_i$, and $b_i$ constitute our node centrality metrics. By studying all three, we compare the graph-theoretical importance of the various brain regions that comprise the connectome, and thus are able to estimate their relative influence within the structural connectome.

## Global topological features

We studied some global topological properties in the networks in order to investigate the balance between integration and segregation, as well as to measure the amount of hierarchical structure, in the different types of connectomes. We estimated the integration simply by means of the global efficiency. The efficiency between two nodes is defined as the inverse of the shortest distance between them [3]. Since ours are weighted graphs, the shortest path is not that which contains the minimum number of edges; instead, the distance associated with each edge was defined as the inverse of its weight, allowing the Dijkstra shortest path [37] $s_{ij}$ to be computed for any given pair of nodes using the NetworkX Python package [36]. The global

efficiency was then computed as the average over the efficiencies between all pairs of nodes

$$E_{glob} = \frac{1}{N(N-1)} \sum_{i \neq j} \frac{1}{s_{ij}} \tag{5}$$

We estimated the degree of segregation in a given network by attempting to partition it into the set of communities for which the value of the modularity is maximum, and then using that maximum value of the modularity as our measure of segregation. The modularity is defined for a certain partitioning of a given network so as to quantify the strength of the connections *within* communities relative to the strength of the connections *between* communities. In the cases of a weighted network with a connectivity matrix $A$, this may be written as

$$Q = \frac{1}{2m} \sum_{i,j} \left[ a_{ij} - \frac{d_i d_j}{2m} \right] \delta(c_i, cj) \tag{6}$$

where $d_i$ is the degree centrality of node $i$ as defined in Eq 1; $c_i$ is the community to which node $i$ belongs; $\delta(u, v)$ is a Kronecker delta, equaling 1 if u = v and 0 otherwise; and $m = \frac{1}{2} \sum_{ij} a_{ij}$. We use the algorithm prescribed in [38] to arrive at the partitioning which maximizes the modularity for each network, and we cite that value of the modularity as an estimate of the degree of segregation. Finally, hierarchical structure in the networks was quantified by means of the global reaching centrality [39], for which computation we also employed the Python package NetworkX [36]. For a graph $G$ with $N$ nodes, GRC may be written as

$$GRC = \frac{\sum_{i \in V} [C_R^{max} - C_R(i)]}{N - 1} \tag{7}$$

where $V$ is the set of nodes in $G$, $C_R(i)$ is the local reaching centrality of node $i$, and $C_R^{max}$ is the maximum value thereof. Local reaching centrality is defined for unweighted directed graphs as the fraction of all nodes in the network that may be reached from node $i$, and generalized for weighted graphs as

$$C_R(i) = \frac{1}{N-1} \sum_{j:0 < n^{out}(i,j) < \infty} \frac{\sum_{k=1}^{n^{out}(i,j)} w_i^{(k)}(j)}{n^{out}(i,j)} \tag{8}$$

where $n^{out}(i, j)$ is the number of links along the shortest path from node $i$ to node $j$, and $w_i^{(k)}(j)$ is the weight of the $k$-th such link.

## Results

We plotted the averaged extended and restricted connectomes in Figs 1 and 2 respectively. In Fig 1, we note very strong edge-weighted connections between the left and right cerebellum cortices and the brainstem, as well as several strong connections in the midbrain and mid-cortex region. With the absence of the brainstem in Fig 2, both sides of the cerebellum appear to be very weakly connected to the rest of the network. Additionally, we no longer see as many strong connections in the midbrain and mid-cortex region. In Fig 3, we isolated the top 20 structures based on node density (degree centrality) and plotted the connectome to show how these structures are connected. The high-ranking structures, which include the brainstem, the thalamus, and the superiorfrontal cortices, seem to form a hub in the midbrain.

The corresponding connectivity matrices for the extended and restricted connectomes are plotted in Fig 4a and 4b respectively. Here we can also see that the extended network is much more strongly interconnected than the restricted one. Interestingly, the interconnectivity between the right and left cortices dramatically drops off in the restricted network, which

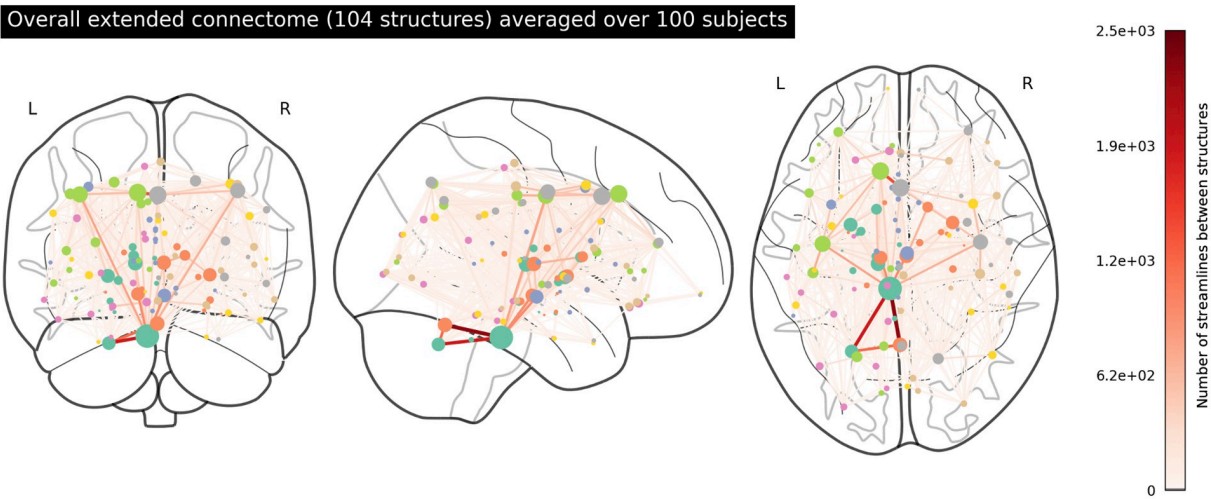

**Fig 1. An overarching connectome featuring the 104 structures in the extended network.** The strength of connections between structures (represented by the coloured nodes) is indicated by the colour, with darker red indicating more streamlines connecting two regions. Node size reflects the raw density of each node. Streamlines correspond to white matter fibre bundles. The full list of the 104 structures in this connectome can be found in Supporting information.

could indicate that some streamlines that connect cortical structures must pass over subcortical structures, and when these structures are eliminated from the calculation of the matrix, these streamlines are missed.

## Centrality

As explained above, we quantified the importance to the network of each node in three different ways; we computed the degree centrality, eigenvector centrality, and betweenness centrality. We present in Fig 5 the 10 highest ranked structures according to each of the three measures, as well as their associated standard deviations, both within extended networks (a)

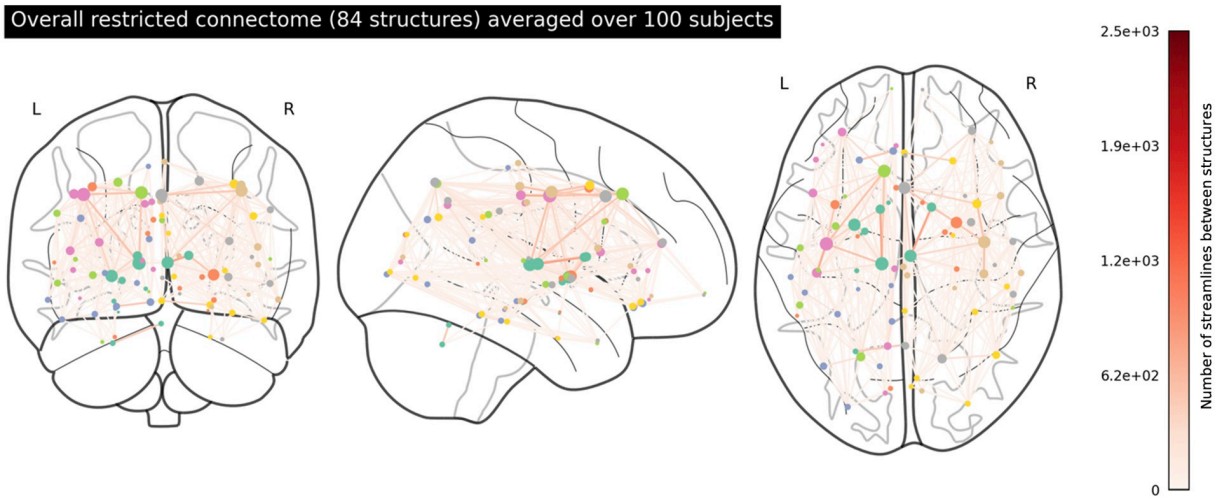

**Fig 2. An overarching connectome featuring the 84 structures in the restricted network.** The full list of the 84 structures in this connectome can be found in S1 Table.

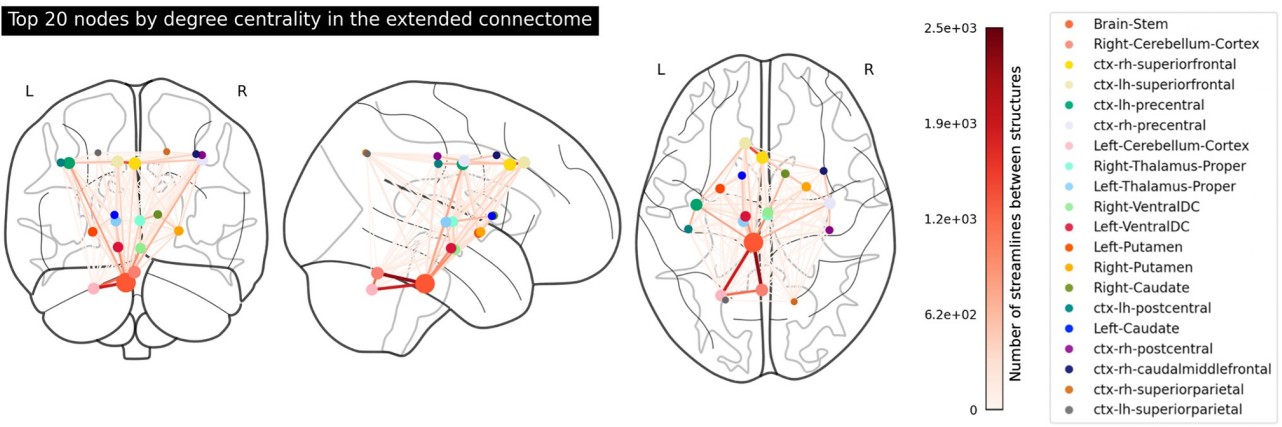

**Fig 3. The top 20 most connected structures in the averaged extended connectome by degree centrality.** The size of the nodes represents the node density (degree centrality) of the structure, and the edge colours represent the number of streamlines connecting two nodes. The structures are listed in the legend in the order of degree centrality.

and restricted ones (b). These were computed by first calculating the centralities for each structure in each connectome, then for each structure averaging across all subjects and computing the inter-subject standard deviations, It is important to note that these bars are not error bars, but standard deviations representing the variability between individuals. and then ranking the structures in descending order of their averages. Each averaged centrality measure was normalized by dividing by its own maximum averaged value, such that the highest ranked structure according to each averaged measure always has a value of unity. In order to take into account the variable size of the structures, we performed the same calculations for normalized

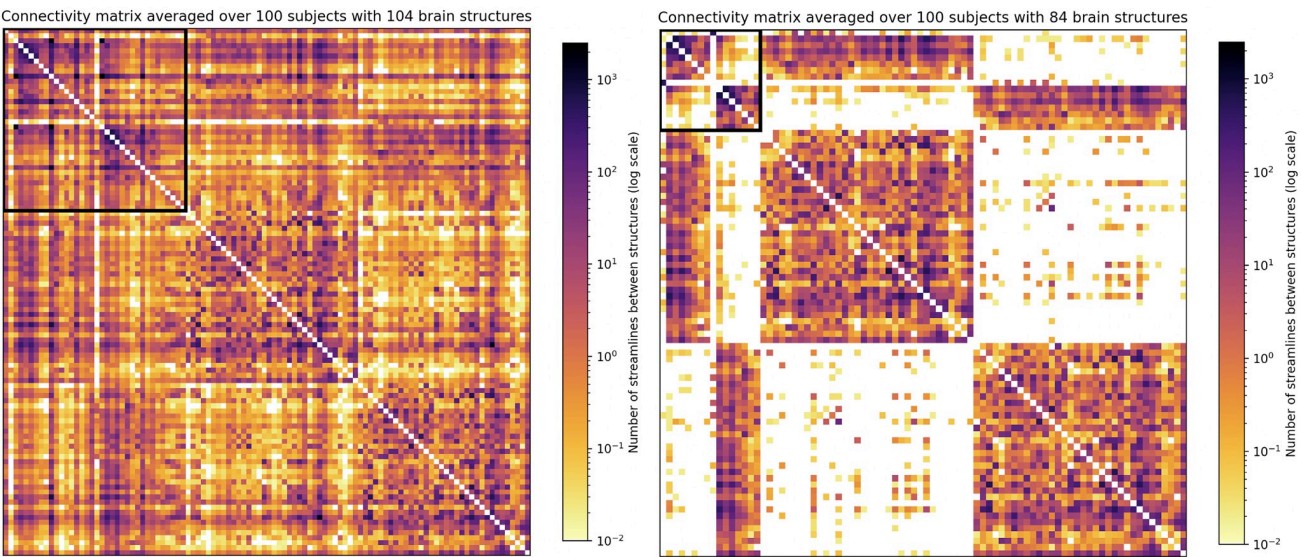

**Fig 4. (a) (left) The structural connectivity matrix for the extended connectome (104 structures), averaged over 100 adult subjects; (b) (right) the structural connectivity matrix for the restricted connectome (84 structures).** The black box indicates subcortical structures. The full list of structures in the extended and restricted connectomes can be found in Supporting information and S1 Table respectively.

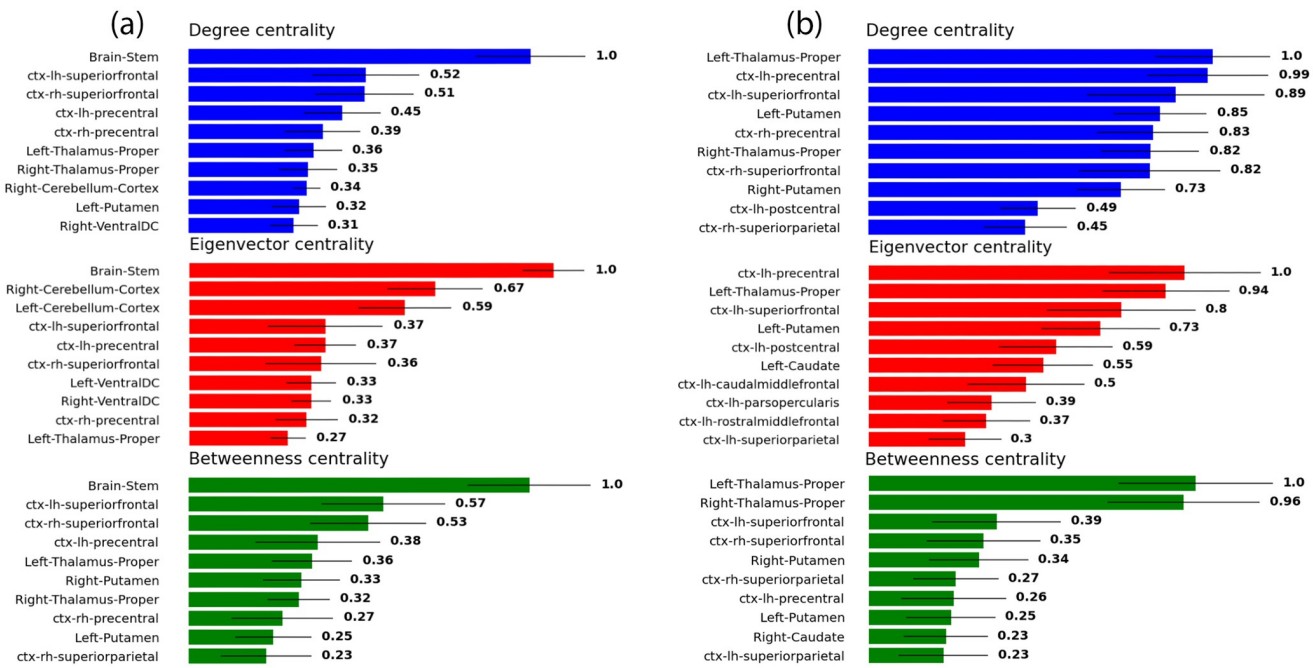

**Fig 5. The top 10 central structures for (a) the extended networks and (b) the restricted networks, according to three different measures: Degree centrality (top), eigenvector centrality (middle), and betweenness centrality (bottom).** Each of the three quantities is normalized with respect to its own maximum value.

versions of each network. The extended and restricted networks were both normalized by dividing the strength of each connection by the sum of the volumes (lists of which are presented in Supporting information and S1 Table) of its two nodes, as in earlier studies [40, 41]. The results for the normalized networks are presented in Fig 6.

We begin by discussing the case of the extended connectome in Fig 5a. In all the three centralities, the brainstem is leading by quite a substantial margin. This result clearly establishes the high centrality of the brainstem within the averaged connectome, at least from the perspective of the three measures considered here. Another structure which displays strong centrality (albeit nowhere near as strong as the brainstem) is the superior frontal cortex, as both the left and right hand-sides feature in the high ranks of all three measures.

The remaining structures featured in Fig 5a, while still displaying considerable centrality, pale in prominence compared to the brainstem; their distributions considerably overlap, as indicated by the bars representing the standard deviations, thereby preventing us from making strong statements as to their precise ranking. These structures include the thalamus and the precentral cortex, and other fairly high-ranking structures such as the ventral diencephalon (DC), the putamen, the cerebellum, and the superior parietal cortex. These structures seem to form a hub in the midbrain as shown in Fig 3.

Fig 5b paints quite a different picture for the restricted connectome. A particularly striking feature, for instance, is the absence of a structure that strongly dominates all others, in the same way that the brainstem does in Fig 5a. In its stead, the most salient subcortical structure appears to be the thalamus, which, while ranking rather highly in general, only appears to eminently dominate in the case of the betweenness centrality. In the cases of degree and eigenvector centralities, there is a notable absence of a single structure that dwarfs the rest of the structures with largely overlapping distributions.

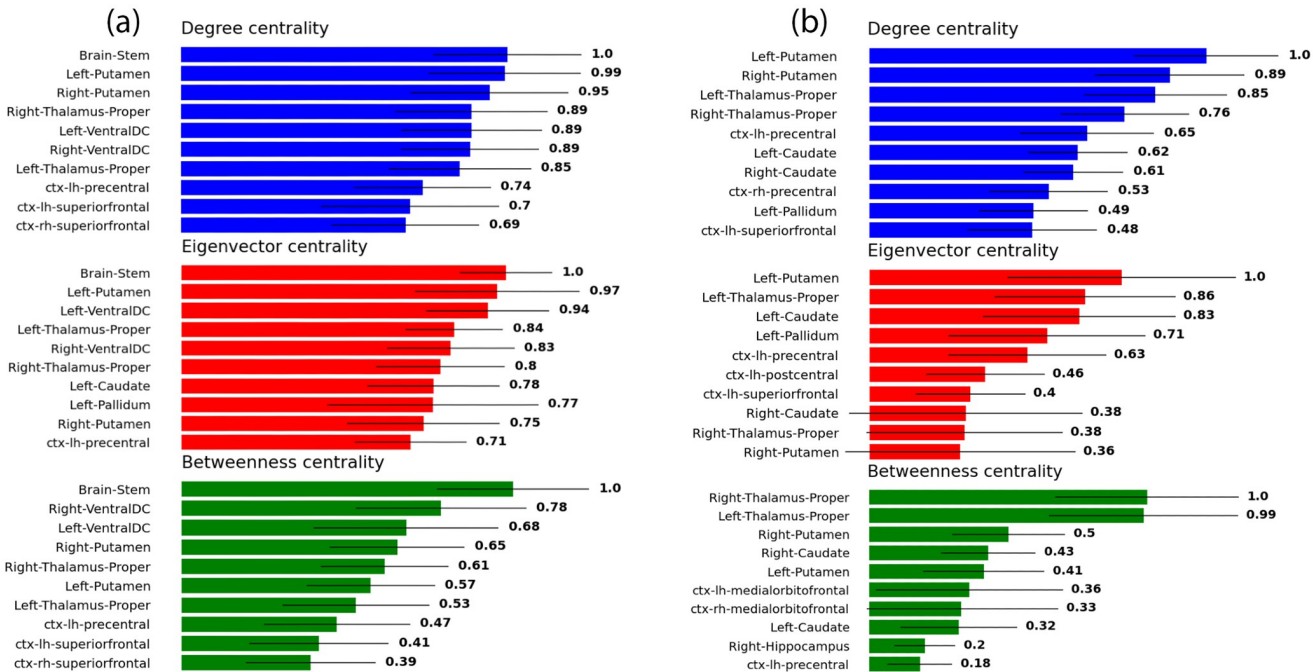

**Fig 6. The top 10 central structures within the *volume-normalized* connectomes, for (a) the extended network and (b) the restricted network, according to three different measures: Degree centrality (top), eigenvector centrality (middle), and betweenness centrality (bottom).** Each of the three quantities is normalized with respect to its own maximum value.

It may also be observed that there exists a difference in the order in which the structures appear between Fig 5a and 5b. It appears that going from the extended connectome to the restricted one, the surviving structures do not retain the order of the mean values of their centralities. The removal of those 20 structures has indeed caused some structures, on average, to rise in centrality, and others to fall. This is also the case for the corresponding normalized networks in Fig 6a and 6b. However, most structures seem to retain their importance in both Figs 5 and 6, when going from the extended network (a) to the corresponding restricted one (b). It may also be perceived that some of the eigenvector and/or betweenness centralities in Fig 6 appear to possess such substantial standard deviations as to broach the zero mark; these, of course, do not suggest that some subjects possess brain regions with negative centrality, but are rather artifacts of the long-tailed subject distributions of centralities for those brain regions.

The cerebellum is another sub-cortical structure which, at first glance, may appear to possess a considerable amount of degree and eigenvector centrality within the extended connectome in Fig 5a. However, when the volume density of streamlines is taken into account, the situation is altered. Consider Fig 6a, which does not feature the cerebellum among any of the high ranking centrality measures. In fact, the means of the degree, eigenvector, and betweenness centralities for the cerebellum in this case were computed to be 0.13, 0.17, and 0 respectively. This strongly suggests that the ostensible high connectivity of the cerebellum implied by Fig 5a may largely be attributable to its sheer size. This is not true of the brainstem, on the other hand, which retains quite a high ranking in Fig 6a across all centrality measures, despite its substantial volume. The superiorfrontal cortex is another example of a structure that seems to lose some prominence when the volume is taken into account, albeit to a much lesser extent than the cerebellum.

## Global topology

Fig 7 presents our analysis of some global topological features in our networks. In (a), we study the balance of integration and segregation in all types of networks; we use the ultimate modularity as a measure of the degree of segregation in a network, which is the maximum modularity that we were able to obtain by employing the algorithm in [38]; and we estimate integration within the networks by means of computing the global efficiency. In (b), we plot the ultimate

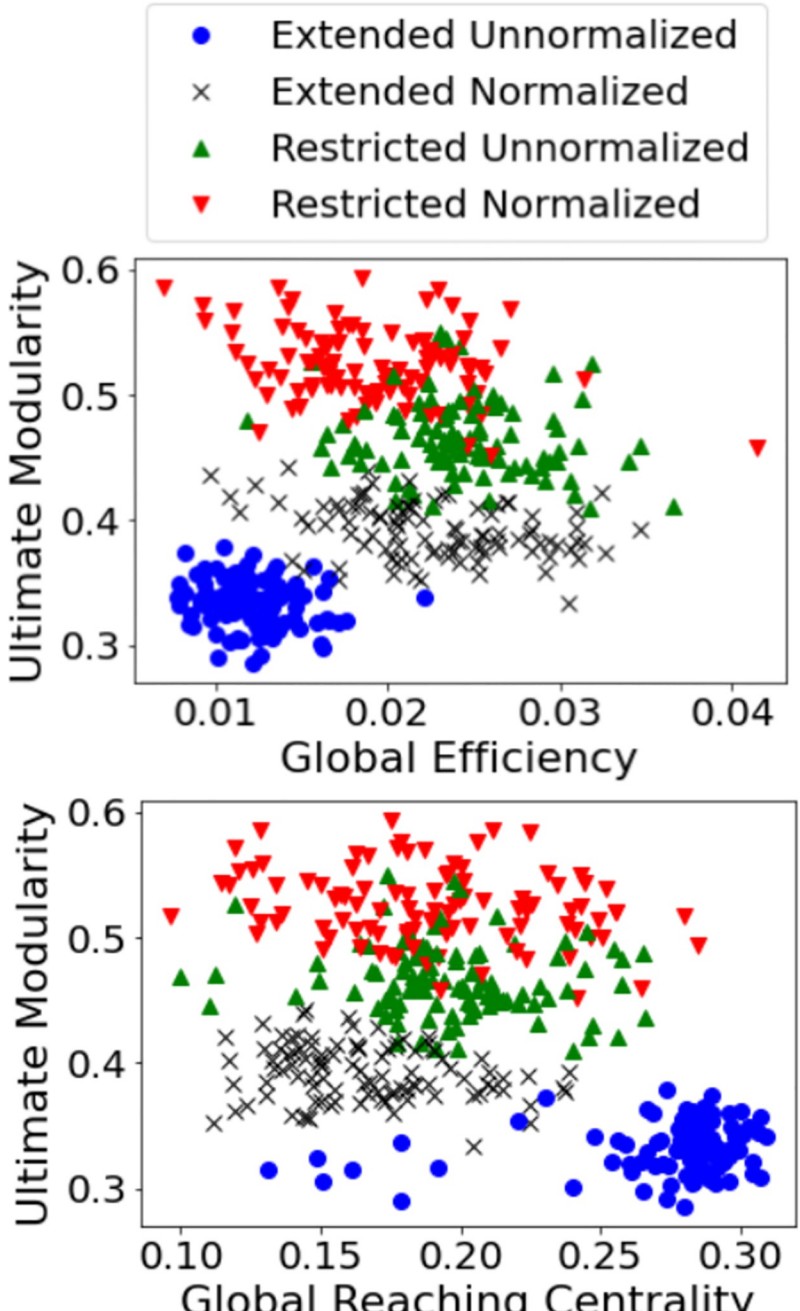

**Fig 7. (a)(top) The balance between integration and segregation in all types of connectomes; (b)(bottom) segregation vs. hierarchy in all types of connectomes.**

modularity against the global reaching centrality, the latter being a measure of the amount of hierarchical architecture within a graph.

It is clear that both plots enjoy considerable separation between the points belonging to different types of connectomes, allowing us to draw conclusions by simple visual inspection of the relative positions of the centers of mass of each set of points. From (a), it appears that restricted connectomes, be they normalized or not, tend to be more segregated on the whole; and that the volume normalization procedure seems to have caused the extended connectomes to become more integrated and more segregated, but the restricted connectomes to become more segregated but rather less integrated. From (b), the global reaching centrality doesn't seem to differ too greatly between the different types of connectomes, with the obvious exception of the extended unnormalized one, which appears to be on average more hierarchical than the rest. These results, as well as those presented above, are discussed in the next section.

## Discussion

Our results unambiguously highlight the structural importance of the brainstem in the extended connectome. All measures of centrality we computed point to the brainstem being the most prominent structure on average within the extended networks, by a margin larger than the differences between the successive structures, and larger than the inter-subject standard deviations. It also became clear upon studying the volume-normalized network that the large size of the brainstem is not solely responsible for its dominance. By contrast, it appears that the high centrality of the cerebellum in the unnormalized network was an artifact of its extensive volume.

Furthermore, the connections of the cerebellum reach primarily to the brainstem, and to very few structures in the midbrain and beyond in the cortex. This becomes clear when comparing the ranking of the cerebellum between Fig 5a, with all 104 structures present, and Fig 5b, with 20 structures (including the brainstem) absent. Upon exclusion of the brainstem, the cerebellum disappears completely from the list of top 10 structures for all three centrality measures. This, in combination with the volume observation, certainly detracts from the importance of the cerebellum within the structural network, which is at least consistent with the accepted idea that the cerebellum does not take any part in conscious processing [42].

As mentioned in the results section, going from the extended to the restricted connectome did not preserve the average centrality order of the surviving structures. While perhaps unsurprising, this observation emphasizes the importance of incorporating as many brain regions as possible when drawing conclusions based on such graph-theoretical considerations. For example, the use of an exclusively cortical atlas, even if only cortical structures are of interest, may yield misleading results.

The global topological measures computed and presented in Fig 7 also reveal some interesting insights. We observed restricted connectomes to exhibit more segregation; this might be attributable to the presence of such dominant structures as the brainstem in the extended connectomes, which, by virtue of their strong connections to many regions in different communities, are bound to detract from the relative importance of intra-community connections. Another observation was the volume normalization procedure's causing the extended connectomes to become more integrated and more segregated; this may seem counter-intuitive at first glance, but it is perfectly possible to simultaneously 1) render a network more relatively well-connected on the whole, and 2) render the communities therein to be more interconnected compared to the rest of the network. The volume normalization procedure, upon its causing the brainstem to be less flagrantly dominant, seems to have had precisely that effect on the extended connectomes. On the other hand, the effect of the same procedure on the restricted

connectomes, which are destitute of such exceptionally dominant structures, seems to be to make them more segregated but less integrated. Finally, the global reaching centrality analysis revealed only the extended connectome unnormalized connectome to be substantially more hierarchical than the rest, on average; this seems to suggest that the brainstem, in all its unnormalized dominance, plays an important role within the hierarchical architecture of the networks, and that that role is diminished when the brainstem loses some of this dominance upon normalization.

The anatomies of the thalamus and of the cortex are believed to be essential to the underlying neural mechanisms [43] for consciousness. When the anatomy of the brain is altered, some disorders of consciousness (DOC), such as a vegetative state (VM) or a minimally conscious state (MCS) are sometimes observed. Indeed, it has been shown that these patients tend to exhibit significantly reduced structural connections between the basal ganglia, thalamus, and frontal cortex [44]. Another study found a significant decrease in volume in the globus pallidus and the thalamus, especially prominent in the thalamus [45]. Recent evidence suggests that the dynamics that underlie loss of consciousness can emerge from randomized neuroanatomical connectivity [46]. The mesocircuit hypothesis proposes that the recovery of consciousness following severe injury that can lead to DOCs is possible, specifically through corticothalamic circuits [47]. The mesocircuit model proposes several structural and dynamic changes that aid in the process of recovery, with structural alterations changing the widespread disconnection of neurons and thus potentially leading to the reestablishment of some learning and memory mechanisms. Recently, it was also found that a loss of consciousness is associated with more structurally-constrained local dynamics and a disruption to structural hubs that contain some subcortical regions [48]. These findings suggest that the specific structural wiring in the brain is important for conscious processing, revealing an important application for the structural connectome.

The high ranking of the brainstem across all examined metrics suggests an important role for the brainstem in information integration. Recent studies suggest that mechanisms in the brainstem are crucial for the constitution of the conscious state [22]. At the very least, the brainstem is responsible for maintaining a most basic level of consciousness, as evidenced by various studies on children with hydranencephaly, who still appeared conscious despite missing most of their cortex [22]. Damage to the brainstem can induce a comatose state, which is the most radical form of disturbance in consciousness [49]. The mesocircuit hypothesis also suggests the brainstem is important for consciousness, positing brainstem/forebrain "arousal systems" that control many thalamic and cortical neurons [47]. Given the evidence linking the brainstem to important aspects of consciousness, and our results suggesting the brainstem is one of the most, if not the most, highly connected anatomical region in the brain, we propose that the brainstem warrants more consideration within notable theories of consciousness. With some theorists suggesting that the upper brainstem may be especially important for maintaining the state of consciousness [20, 21], this also calls for more structural network analyses with finer parcellations of the brainstem. Our analysis was limited to the brainstem as a whole, due to the nature of the parcellation included in the HCP data. The HCP parcellation, which was built into the preprocessed data that we used, is rather broad and does not parcellate known regions like the brainstem or the thalamus into subregions like the medulla, pons, or thalamic nuclei.

In the context of consciousness, two main theories could be interpreted with our results. Global workspace theory is one such theory which proposes a fleeting memory capacity enabling access between brain functions [50]. While it is a functional theory, it does draw on a structurally inspired framework [51], and it is thought that the thalamus and cortex work as an integrated system to create this global access network, because they are functionally integrated

to a degree where they constitute a single functional system [52]. Our metrics suggest that the superior frontal and precentral cortices are strongly connected to the rest of the network, as are the right and left thalami, which suggests that a strong structural cortico-thalamic network may be present. Integrated Information Theory (IIT) defines consciousness as the capacity of a system to integrate information [53], which is quantified by $\phi_{max}$. To achieve a high $\phi_{max}$, a structure must be sufficiently functionally integrated [54]. There is evidence to suggest a correlation between structural and functional connectivity, and given that the physical substrates of consciousness (PSC) can split as a result of anatomical disconnections [55], there is reason to believe that structural connectivity does play a fundamental role in IIT. Examining our results in the framework of IIT, under the assumption that structural and functional connectivity are related, we see a likelihood for the brainstem to give rise to a high $\phi_{max}$ given that it is the top-scoring structure across all our structural metrics. This provides further motivation to examine the brainstem from a functional perspective in the context of IIT as well. We acknowledge, however, that our results cannot provide a complete picture of the role of structural connectomics in the generation and/or perpetuation of conscious processes, since we have only considered healthy patients. It could be the case that the regions of high connectivity that we have identified, such as the brainstem and the thalamus, are related to processes other than consciousness. It is necessary, then, to conduct the same network analysis with DOC patients as well in order to gauge the extent to which high structural connectivity plays a role in consciousness. This is an interesting question for future research.

Finally, we make a connection between the results presented here and the newly emerging ideas involving quantum effects in the brain (see, for instance, [56]). Given that the delivery of anaesthetics specifically targeting the brainstem has been shown to be sufficient for the induction of loss of consciousness [57], and given the recent proposal that quantum effects may be important for xenon-induced anaesthesia [58], the prominent role enjoyed by the brainstem in our structural analysis may have potential implications for quantum-mechanical explanations of consciousness.

There are several limitations with our computational fibre tracking method. One of the largest limitations with fibre tractography using diffusion tensor imaging (DTI) is that the tensor model, which characterizes the orientation of white matter fibres, cannot differentiate complex configurations of crossing fibres in a single voxel [59]. Although the incorporation of constrained spherical deconvolution (CSD) in our computation improves this error, there are a few other advanced computational methods available, such as Q-ball imaging and diffusion spectrum imaging, and there is no general consensus on which method is best [59]. The CSD uses an empirically derived response function, which estimates the direction of fibre tracts and assumes that white matter fibres do not differ in their diffusion properties between tracts [60]. A common criticism of this approach, however, is that the response function may not be constant throughout the brain [60], which may alter the measured distribution of fibres within a voxel. Additionally, while there is no clear consensus on which parcellation method is best, the structures extracted from the HCP structural images used in this paper do not allow for a more detailed examination of the brainstem, as it has not been parcellated into smaller regions such as the upper nuclei, medulla, and pons.

The CSD approach is most notably used in MRtrix, a widely-used software pipeline for streamline tractography and connectome generation. It is formulated such that either a probabilistic or deterministic tracking algorithm can be used to track and generate streamlines. The tckgen function uses a probabilistic tracking algorithm by default, taking a fibre orientation distribution (FOD) image as input, where streamlines are considered more probable to follow a path where the FOD amplitudes along that path are also large. Similarly, our Python approach uses CSD to generate an FOD that guides the ProbabilisticDirectionGetter in the

DIPY module, producing tracking direction seeds that are then used in the tracking. The difference between the methods, however, lies in their approaches to track termination. MRtrix terminates a track during the tracking process if no suitable FOD peak can be found whose amplitude is above a user-specified threshold [61]. In Python, we first used a Constant Solid Angle (CSA) Orientation Distribution Function (ODF) model, or the CsaOdfModel function, to estimate the orientation of tract segments at a point in the image. Then, the CSA/ODF model was passed into the `ThresholdStoppingCriterion` function, which determines the stopping criterion for the tracks based on the peaks of the ODF. Since the Desikan-Killiany atlas, which produces the restricted connectome, is available for use with MRtrix, we ran the software on one subject to compare the two methods. We noticed some quantitative differences, as some structures appeared to be higher or lower scoring in the MRtrix version than in our Python version, but qualitatively, the matrices were similar. Largely the same structures appeared in the top ten across all our centrality measures. An additional key difference between the two methods is that MRtrix allows for the manual setting of the number of streamlines desired, whereas in DIPY, the number is determined empirically based on a subject's structural data. This renders it difficult to conduct a direct quantitative comparison between the two methods, since the convergence of the MRtrix results as a function of the number of streamlines may be more ambiguous. Nevertheless, our qualitative comparison shows there can be differences in the strength of certain structures depending on the computational methods used. An extension of our work could involve computing the same centrality measures for an MRtrix-generated connectome if an atlas including both cortical structures and subcortical structures (including the brainstem) should become available, and to quantify the optimal number of streamlines in MRtrix.

Our computed centrality measures, while powerful tools, also come with some limitations. Degree centrality, for instance, straightforward as it is to compute, assumes that the importance of a node is determined entirely by the amount of connections it has to other nodes in the network. Hence, such a measure could in principle overestimate the importance of a node which is connected to a large number of irrelevant nodes. To rectify this shortcoming, eigenvector centrality gives precedence to those nodes that are connected to other nodes with high centrality, as opposed to considering all connections on equal grounds. One possible limitation of eigenvector centrality is that it says little of the difference between the degree of a node and those of its neighbours, which is an important consideration for the study of assortativity and modularity of networks [62].

These two measures are complemented by betweenness centrality, which identifies the most important nodes of a network as those which lie in the most strategic locations with respect to the shortest paths between pairs of nodes. While certainly a strong measure, it suffers from its own limitations, such as a computational cost that quickly becomes unfeasible as a function of network size. Although our moderately sized networks allowed us to largely evade this limitation in this work, it is bound to present a challenge for future studies considering significantly larger brain networks. Furthermore, betweenness centrality is implicitly based on the assumption that information propagates through the network serially along the most efficient routes. This has been argued to be a flawed assumption when considering brain networks, as it presupposes that individual components possess global information about the network topology [63], a condition that is unlikely to be satisfied in the context of biological systems [64].

The notion of network directionality constitutes another limitation within our approach. Graph-theoretical studies based on neuroimaging data typically represent connections within networks as undirected, owing to the inability of such non-invasive techniques to resolve the directionality of anatomical connections [19]. Our investigation presented here is no

exception. However, it is believed that brain network connections are inherently directional in nature, and that the assumption of undirected connections introduces inaccuracies to the computation of topological properties, especially to the process of hub identification [65]. Hence, results obtained here may in principle be improved in future studies by constructing directed networks. Currently, some invasive techniques are capable of resolving axonal directionality of non-human connectomes [64].

A natural extension of this research would be to examine the role of the brainstem in more detail, ideally with a finer parcellation of the entire connectome. Since our goal was to examine how the brainstem generally fits into the structural connectome, we did not parcellate it further, but a finer parcellation could alter the network even more and may shed light on where the high structural connectivity that we see in the brainstem is concentrated. The brainstem could be parcellated into the midbrain, pons, medulla, and superior cerebellar peduncles [66]. Another topic for future investigation is establishing connections with functional observations by means of implementing a dynamical model in the context of our network. A number of such models have been applied to brain networks to assess structure-function relationships, such as neural mass models [67, 68], models of Wilson-Cowan oscillators [41, 69], and spin models [32–34]. Studying critical phenomena and phase transitions within these systems allows for the possibility of making connections with the critical brain hypothesis [70, 71].

## Conclusion

We examined structural connectivity across 100 healthy adult subjects, using graph theory tools and structural brain networks that include the brainstem. We found that the brainstem scores the highest in all our computed centrality measures, followed by the superiorfrontal cortices and the right and left thalami. When the brainstem and several other subcortical structures are removed in the restricted connectome, we observed that the connections between the surviving regions were substantially altered, and we found that the most prominent structures revealed by the centrality analysis were rather different on average. This stresses the importance of incorporating all brain regions in structural network analyses, and suggests that the use of a non-comprehensive list of structures may give rise to misleading conclusions. Overall, the brainstem certainly merits more consideration in structural and functional analyses, and potentially also in theories of consciousness.

## Supporting information

**S1 Table. A list of the 104 brain structures used in the extended connectivity matrix, with the corresponding number of streamlines attached to each structure and the average volume of the structure across 100 subjects.**
(PDF)

**S2 Table. A list of the 84 brain structures used in the restricted connectivity matrix, with the corresponding number of streamlines attached to each structure.**
(PDF)

**S3 Table. The list of 20 structures removed in the restricted connectivity matrix.** This matrix corresponds with the standard MRtrix parcellation scheme, widely used in studies of brain networks. The list of cortical structures remained the same.
(PDF)

## Acknowledgments

We would like to thank Emma Towlson and Wilten Nicola for their valuable input, guidance, and discussion, as well as Joern Davidsen, Davor Curic, and Omid Khajehdehi from the Complexity Science Group (CSG). Additionally, we thank the HCP for providing access to their data.

## Author Contributions

**Conceptualization:** Salma Salhi, Youssef Kora, Hadi Zadeh Haghighi, Christoph Simon.

**Data curation:** Salma Salhi.

**Formal analysis:** Youssef Kora.

**Funding acquisition:** Christoph Simon.

**Investigation:** Salma Salhi, Youssef Kora.

**Methodology:** Salma Salhi, Youssef Kora, Gisu Ham.

**Project administration:** Youssef Kora, Christoph Simon.

**Software:** Salma Salhi, Youssef Kora, Gisu Ham.

**Supervision:** Christoph Simon.

**Validation:** Salma Salhi, Youssef Kora, Christoph Simon.

**Visualization:** Salma Salhi, Youssef Kora.

**Writing – original draft:** Salma Salhi, Youssef Kora.

**Writing – review & editing:** Salma Salhi, Youssef Kora, Hadi Zadeh Haghighi, Christoph Simon.

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
