## [Decision Letter · Decision Letter 0]

12 Dec 2022

PONE-D-22-20864Network analysis of the human structural connectome including the brainstem: a new perspective on consciousnessPLOS ONE

Dear Dr. Salhi,

Thank you for submitting your manuscript to PLOS ONE. After careful consideration, we feel that it has merit but does not fully meet PLOS ONE’s publication criteria as it currently stands. Therefore, we invite you to submit a revised version of the manuscript that addresses the points raised during the review process.

The reviewers listed a number of comments related to the methodology adopted during the analysis of the dataset. You should make clear every preprocessing step that will further enhance the readability and the potential

repeatability of the your study.

We look forward to receiving your revised manuscript.

Kind regards,

Stavros I. Dimitriadis

Academic Editor

PLOS ONE

https://journals.plos.org/plosone/s/fileid=ba62/PLOSOne_formatting_sample_title_authors_affiliations.pdf.

2. Thank you for including your ethics statement:  "Consent was not obtained as the data is publicly available as part of the Human Connectome Project. The data were analyzed anonymously".  

(1). For studies reporting research involving human participants, PLOS ONE requires authors to confirm that this specific study was reviewed and approved by an institutional review board (ethics committee) before the study began. Please provide the specific name of the ethics committee/IRB that approved your study, or explain why you did not seek approval in this case.

(2). Please provide additional details regarding participant consent. In the ethics statement in the Methods and online submission information, please ensure that you have specified (1) whether consent was informed and (2) what type you obtained (for instance, written or verbal, and if verbal, how it was documented and witnessed). If your study included minors, state whether you obtained consent from parents or guardians. If the need for consent was waived by the ethics committee, please include this information.

3. Please expand the acronym “S.S., Y.K., G.H., C.S., H.Z-H.” (as indicated in your financial disclosure) so that it states the name of your funders in full.

“This work was supported by the Natural Sciences and Engineering Research Council (NSERC) 427 of Canada. We would also like to thank Emma Towlson and Wilten Nicola for their valuable 428 input, guidance, and discussion, as well as Joern Davidsen, Davor Curic, and Omid Khajehdehi 429 from the Complexity Science Group (CSG). Additionally, we thank the HCP for providing 430 access to their data.”

“All authors (S.S., Y.K., G.H., C.S., H.Z-H.) were funded by the NSERC Discovery Grant RGPIN-2020-03945, URL: https://www.nserc-crsng.gc.ca/index_eng.asp. The funders had no role in study design, data collection and analysis, decision to publish, or preparation of the manuscript.”

Additional Editor Comments:

Dear authors

The authors listed a number of comments related to the methodology adopted during the analysis of the dataset.

You should make clear every preprocessing step that will further enhance the readability and the potential

repeatability of the your study.

Reviewers' comments:

Reviewer's Responses to Questions

**Comments to the Author**

1. Is the manuscript technically sound, and do the data support the conclusions?

Reviewer #1: Partly

Reviewer #2: Yes

2. Has the statistical analysis been performed appropriately and rigorously? 

Reviewer #1: I Don't Know

Reviewer #2: No

3. Have the authors made all data underlying the findings in their manuscript fully available?

Reviewer #1: Yes

Reviewer #2: Yes

4. Is the manuscript presented in an intelligible fashion and written in standard English?

Reviewer #1: Yes

Reviewer #2: Yes

5. Review Comments to the Author

Reviewer #1: The work by Salhi and coworkers analyzes, in brief, how the topological network properties of the human connectome change when the brainstem structures are included as part of the network. The role of these structures as hubs for the topological connections are immediately spotlighted, with the use of several local topological measures. This make the authors to hypothesize about how these findings could inform theories about consciousness.

The work is clearly presented, and the results sustain the main conclusions of the work. However, I find the work to be unambitious in the analysis that could be performed and too ambitious in the conclusions. The results are not surprising given what is already know about the brainstem, however the authors try to give an impression of a game-changing discovery about consciousness. Although the role of hubs in maintain/supporting states of consciousness is well reported, an analysis based only on graph theory may not be enough to make the link here. I think that a few extra analyses could be easily added, giving more weight to the results and conclusions.

Some suggestions for extra analysis may include: 1) Relate the structural connectivity findings with the functional (fMRI) data. I understand that this is available from HCP, too. 2) Do some simulations with dynamical models. I understand that there is a difficulty here, as the typical dynamics of cortical areas may not be suited for subcortical structures (although some works do it for the thalamus). But some very generic model may do the work. 3) Incorporate some global topological features in the analysis, such as global efficiency to measure integration, or clustering coefficient/modularity to measure segregation. Is the integration/segregation balance similar between the "restricted" and "extended" connectomes? If not, why? Rich-club analysis and s-core decomposition are other options. There may be nothing relevant there, but it would be still interesting to note it. 4) Report data dispersion: how variable are the results from one individual to another?

Specific/minor comments:

Introduction: (line 28) “…, it has been shown that the strength and persistence of functional connectivity is constrained by the anatomical cortex”. The FC is constrained by SC, but it is important to remark that other "players" contributes in shaping brain's FC: local dynamics, E/I balance, neuromodulators, etc.

Introduction: When talking about DOCs, I suggest to include this reference (introduction and/or discussion): https://www.nature.com/articles/s42003-022-03330-y

Second half of Introduction may be restructured a little bit. There are some unnecessary methodological details and the end is too abrupt. The main findings should be given at the end of Intro, too.

Conclusion, line 406: "using a specialized computational approach" is vague. Just write "using graph theory tools and structural brain networks that includes the brainstem."

Figures 1 and 2: Increase the fontsize of the color bar. Add a color bar label.

Figures 3: indicate x and y axis labels (ROIs, brain regions, etc). Also a log scale in the plot (or plotting log(weight)) may help to see better some contrasts.

Figure 4 and 5: Just for visualization purposes, I suggest (in addition to the bars) to plot brain regions projected in the brain. The authors can use colors or modify ROIs sizes (radious of the spheres) in accordance to the nodal measures. For example, Brainstem can be plotted 3x the size of Left-Putamen. Also, it could be great if the authors can add a superior title to Figures 4 and 5, for differentiating between them.

Reviewer #2: Author have studied how brainstem impact the structural brain connectivity using helth control DTI matrix from open (HCP) database. It is very interesting to see with and without brainstem, structural connectivity changes the important hub region. Even though author used only healthy subject to infer consciousness processes, author have addressed brainstem function for consciousness. I have below technical, result representation and interpretation issues, which need to be addressed.

For Introduction:

- Author could discusses recent paper by López-González et al (2021), as López-González et al. shows disruption of the structural connectivity using DTI (a similar approach which author have used in this study) in disorder of consciousness patients, specifically to subcortical areas.

For Method:

- Line 147: How many maximum number of stream line count was used in the diffusion affine matrix?

- Line 164: I think author have used absolute graph measures (i.e., betweenness centrality). It would be interesting to see the normalized betweenness centrality. See Holla et al, 2017 to extract normalized graph measures.

- Did the author compute the betweenness centrality for different threshold (using density/sparsity based thrsholding). Author should use range of thrsholding (i.e., Sparsity fro 0.01 to 0.5) in the structural connectivity matrix and compute the betweenness centrality in all different thresholder structural connectivity matrix to show the results are not by chance. (See Holla et al, 2017).

- Author should provide detail of the used atlas (i.e., extended connectome atlas) in the method section. Probably the paragraph of introduction line 90-96, could move to method section.

- Which part of the brainstem have included in the atlas? Is it whole brainstem or any specific part (e.g, midbrain). Author could provide detail of chosen brainstem areas/atlas and why they chosen that.

For Results:

- What is the bar graph represented in the figure 4 & 5, is it mean values? Also for the supplementary table S1 & S2. Author should provide mean and standard deviation values. The bara graph (figure 4 & 5) is hides the results, author should use dotted violin plot which could show subject variability.

- Did the author used any statistics for graph measures (figure 4 & 5), to see if the hub regions (i.e., top 10 brain regions) significantly different by with and without brain steam connections for the structural connectivity?

For Discussion

- Line 272: “….brainstem warrants more consideration within notable theories 272 of consciousness” – its not true. The mesocircuit hypothesis (Schiff, 2010) suggest brainstem is link to consciousness. Indeed, author did not discusses “mesocircuit hypothesis” which is an important conscious theory which suggest brainstem-thalamus and cortical circuits play an important role in consciousness process. This theory should be discussed.

- Author discussed brainstem as a whole. However brainstem parts are more interesting in the respect of consciousness. Author should address, if there is specific part of brainstem (e.g, midbrain) is associate with conscious process.

- The author used only healthy subject at awake state not in any alter conscious state (e.g., DOC patients), so I think to infer that a region have higher connectivity in healthy subject relay to conscious state/process is not appropriate. Instead of consciousness, it could be any functions.

Important suggestion: Author should provide the code used and structural connectivity data during or after acceptance of the paper, so some one can replicate the study. Especial the structural connectivity matrix as the author used open data base. This will not only help to reproduce the results also researcher could use structural matrix as a template for functional MRI studies.

Reference:

López-González, A. et al (2021). Loss of consciousness reduces the stability of brain hubs and the heterogeneity of brain dynamics. Communications biology, 4(1), 1-15.

Holla, B. et al. (2017). Disrupted resting brain graph measures in individuals at high risk for alcoholism. Psychiatry Research: Neuroimaging, 265, 54-64.

Schiff, N. D. (2010). Recovery of consciousness after brain injury: a mesocircuit hypothesis. Trends in neurosciences, 33(1), 1-9.

6. PLOS authors have the option to publish the peer review history of their article (what does this mean?). If published, this will include your full peer review and any attached files.

Reviewer #1: **Yes: **Patricio Orio

Reviewer #2: **Yes: **Rajanikant Panda

---

## [Author Response · Author response to Decision Letter 0]

6 Feb 2023

We thank both reviewers for their fair and thorough review of our manuscript. We have strived to accommodate their comments and recommendations, which we address below.

Response to Reviewer #1

General comments: “The work is clearly presented, and the results sustain the main conclusions of the work. However, I find the work to be unambitious in the analysis that could be performed and too ambitious in the conclusions. The results are not surprising given what is already know about the brainstem, however the authors try to give an impression of a game-changing discovery about consciousness. Although the role of hubs in maintain/supporting states of consciousness is well reported, an analysis based only on graph theory may not be enough to make the link here. I think that a few extra analyses could be easily added, giving more weight to the results and conclusions.”

We thank the reviewer for their insightful recommendations. The response below addresses the concern about including more analyses. We have also changed some aspects of the introduction and the discussion to reduce the emphasis on consciousness and avoid implying any discoveries on consciousness. In the discussion, we relate our findings to two theories of consciousness in brief, and emphasize that our results cannot provide a full picture of the role of structural connectivity in consciousness because we have not included patients with disorders of consciousness in our analyses. 

Reviewer #1 states: “Some suggestions for extra analysis may include: 1) Relate the structural connectivity findings with the functional (fMRI) data. I understand that this is available from HCP, too. 2) Do some simulations with dynamical models. I understand that there is a difficulty here, as the typical dynamics of cortical areas may not be suited for subcortical structures (although some works do it for the thalamus). But some very generic model may do the work.”

We appreciate the Reviewer’s suggestion to incorporate dynamical modeling and connections to functional data. Indeed, we have conducted a research investigation in which we employed a dynamical model to investigate structure-functional relationships in the human brain, but have elected to communicate the results in a separate work, recently submitted to the arXiv and found here https://arxiv.org/abs/2301.01243. Therein we have explored how the underlying structure affects some functional behavior such as global excitation within the Wilson-Cowan model. We have not, however, compared directly with empirical fMRI data, which is something that could be a topic for a future investigation. 

Reviewer #1 states: “3) Incorporate some global topological features in the analysis, such as global efficiency to measure integration, or clustering coefficient/modularity to measure segregation. Is the integration/segregation balance similar between the "restricted" and "extended" connectomes?”

In an effort to accommodate the Reviewer’s suggestion, we have incorporated, following the centrality analysis, an analysis of global topological features in which we computed such quantities as the global efficiency and modularity for 6 subjects, in order to measure and compare integration and segregation between extended and restricted connectomes. The results are presented in Fig. 7 in the new manuscript. We found the integration-segregation balance of the different sorts of connectomes to differ substantially, the most salient observation being that the dominant presence of the brainstem seems to cause the networks to be both less integrated and less segregated on the whole.

4) Report data dispersion: how variable are the results from one individual to another?

We thank the Reviewer for this suggestion, and agree that some presentation of the variability of the results between individuals is worthwhile. To that end, we have computed for each structure in each connectome the standard deviations corresponding to individual variability, and represented them as error bars in Figs 5 and 6. 

Reviewer #1 states: “Introduction: (line 28) “…, it has been shown that the strength and persistence of functional connectivity is constrained by the anatomical cortex”. The FC is constrained by SC, but it is important to remark that other "players" contributes in shaping brain's FC: local dynamics, E/I balance, neuromodulators, etc.”

We have amended the line to the following: “... it has been shown that the strength and persistence of functional connectivity is constrained by the anatomical cortex [15], amongst other key factors such as local dynamics [16] and neuromodulation [17]” to ensure that other factors that shape the FC are mentioned. References 16 and 17 have been introduced to support this. 

Reviewer #1 states: “When talking about DOCs, I suggest to include this reference (introduction and/or discussion): https://www.nature.com/articles/s42003-022-03330-y “

We thank the reviewer for bringing this paper to our attention and have cited it in our discussion of DOCs in the discussion, in the sentence “Recent evidence suggests that the dynamics that underlie loss of consciousness can emerge from randomized neuroanatomical 

connectivity [46]”. 

Reviewer #1 states: “Second half of Introduction may be restructured a little bit. There are some unnecessary methodological details and the end is too abrupt. The main findings should be given at the end of Intro, too.

We think that this is a fair assessment on the part of the Reviewer, and have decided to remove the extra methodological details. We have also provided a summary of our main findings at the end of the introduction. 

Reviewer #1 states: “Conclusion, line 406: "using a specialized computational approach" is vague. Just write "using graph theory tools and structural brain networks that includes the brainstem."”

We have made the change in compliance with the Reviewer’s request. 

Reviewer #1 states: “Figures 1 and 2: Increase the fontsize of the color bar. Add a color bar label.”

We have increased the fontsize of the colour bar, and additionally have elected to change the size of the nodes such that they are reflective of the degree centrality of each node. This is so that the graphs show both the strength of the edges and the nodes. 

Reviewer #1 states: “Figures 3: indicate x and y axis labels (ROIs, brain regions, etc). Also a log scale in the plot (or plotting log(weight)) may help to see better some contrasts.”

We thank the reviewer for these suggestions and have changed both figures 3a and 3b (now 4a and 4b, since we added a new figure above this one) to a log scale. We think that adding labels for each structure would make the plot look cluttered, but we have elected to instead draw boxes around the subcortical structures so that this ROI is easier to make out. 

Reviewer #1 states: “Figure 4 and 5: Just for visualization purposes, I suggest (in addition to the bars) to plot brain regions projected in the brain. The authors can use colors or modify ROIs sizes (radious of the spheres) in accordance to the nodal measures. For example, Brainstem can be plotted 3x the size of Left-Putamen. Also, it could be great if the authors can add a superior title to Figures 4 and 5, for differentiating between them.

We thank the Reviewer for this insightful suggestion, which has inspired us to make the following visualization changes: we changed figures 1 and 2 such that the sizes of the nodes are reflective of their respective degree centralities. However, rather than make direct changes to figures 4 and 5 (now 5 and 6) for the purpose of visualizing the main hubs, we have elected to add an additional figure (figure 3) that visualizes the top 20 structures based on degree centrality.

Response to Reviewer #2

General comments: “Author have studied how brainstem impact the structural brain connectivity using helth control DTI matrix from open (HCP) database. It is very interesting to see with and without brainstem, structural connectivity changes the important hub region.”

We thank the reviewer for their positive comments and insightful suggestions. 

Reviewer #2 states: “Even though author used only healthy subject to infer consciousness processes, author have addressed brainstem function for consciousness.”

We have restructured and rewritten parts of the introduction and discussion to include less emphasis on consciousness. We also changed the title of the manuscript from “Network analysis of the human structural connectome including the brainstem: a new perspective on consciousness” to “Network analysis of the human structural connectome including the brainstem”. 

Reviewer #2 states: “Author could discusses recent paper by López-González et al (2021), as López-González et al. shows disruption of the structural connectivity using DTI (a similar approach which author have used in this study) in disorder of consciousness patients, specifically to subcortical areas.”

We thank the reviewer for bringing this article to our attention and have cited it in the introduction.

Reviewer #2 states: "How many maximum number of stream line count was used in the diffusion affine matrix?”

The following statement was added to the Materials and Methods section to explain the lack of a maximum streamline count: “Note that this function does not contain a parameter specifying the maximum number of streamlines, but instead only has a parameter specifying the maximum number of steps taken to generate streamlines (to prevent infinite loops) which is 500 steps by default. The function also uses the direction getter, the affine matrix, the seeds, and the stopping criterion to generate the streamlines. For each subject, the total number of streamlines is thus on the order of 10^4.” 

Reviewer #2 states: “I think author have used absolute graph measures (i.e., betweenness centrality). It would be interesting to see the normalized betweenness centrality. See Holla et al, 2017 to extract normalized graph measures.” “

To begin with, we normalized our betweenness centrality by dividing by the maximum possible betweenness centrality of a node in a network, i.e., n(n-1)/2. We also further normalized the list of betweenness centralities for each structure by dividing by the maximum value thereof, such that the highest ranking structure always has a value of unity. By taking a look at the work by Holla et al, we perceive that there is a notion of normalization therein that involves dividing the value of certain global properties of the network, like the clustering coefficient and the characteristic path length, by the corresponding averaged value for 1000 randomized networks with the same number of nodes, edges, and degree distribution. But it is not clear to us how that applies to the betweenness centrality, which is a quantity computed for each node separately, and which we already normalized in two different senses as explained above.

Reviewer #2 states: “Did the author compute the betweenness centrality for different threshold (using density/sparsity based thrsholding). Author should use range of thrsholding (i.e., Sparsity fro 0.01 to 0.5) in the structural connectivity matrix and compute the betweenness centrality in all different thresholder structural connectivity matrix to show the results are not by chance. (See Holla et al, 2017).”

We have also elected not to employ a thresholding method which would discard some edges; instead, all edges were taken into consideration and distances and shortest paths were calculated on that basis. Our understanding is that thresholding is primarily utilized to simplify graphs that are very large and very dense in an effort to reduce the computational complexity of the calculation. In our case, however, we had the luxury of being able to perform all computations without the need to remove any edges. 

Reviewer #2 states: “Author should provide detail of the used atlas (i.e., extended connectome atlas) in the method section. Probably the paragraph of introduction line 90-96, could move to method section.”

We agree with the Reviewer and have decided to move the paragraph beginning with “MRTrix is perhaps the most widely used software in brain network studies…” into the methods section. 

Reviewer #2 states: “What is the bar graph represented in the figure 4 & 5, is it mean values? Also for the supplementary table S1 & S2. Author should provide mean and standard deviation values. The bara graph (figure 4 & 5) is hides the results, author should use dotted violin plot which could show subject variability.”

The Reviewer is right; the bar graphs in figure 4 and 5 (now 5 and 6) do indeed represent mean values. We also agree with the Reviewer that some presentation of individual subject variability is warranted. Therefore we have computed such standard deviations and represented them as bars centered around the averages in the figures, which we determined to be the cleanest way of doing so. 

Reviewer #2 states: “Did the author used any statistics for graph measures (figure 4 & 5), to see if the hub regions (i.e., top 10 brain regions) significantly different by with and without brain steam connections for the structural connectivity?”

Following from the variabilities we computed, one can see the distribution of the centralities for the top brain regions. Besides the brainstem’s being decidedly dominant, the distributions of the other structures considerably overlap, and therefore we cannot make strong statements as to their precise ranking with and without the brainstem. We have made a statement in the results section to that effect, namely “The remaining structures featured in Fig 5a, while still displaying considerable centrality, pale in prominence compared to the brainstem; their distributions considerably overlap, as indicated by the bars representing the standard deviations, thereby preventing us from making strong statements as to their precise rankings”.

Reviewer #2 states: “Line 272: “….brainstem warrants more consideration within notable theories 272 of consciousness” – its not true. The mesocircuit hypothesis (Schiff, 2010) suggest brainstem is link to consciousness. Indeed, author did not discusses “mesocircuit hypothesis” which is an important conscious theory which suggest brainstem-thalamus and cortical circuits play an important role in consciousness process. This theory should be discussed.”

We thank the reviewer for pointing us to this theory. We mentioned this theory in the discussion, specifically when commenting on the relationship between functional and structural connectivity in the context of consciousness, since the mesocircuit model states that structural alterations can lead to functional changes as well. Additionally, we mentioned the mesocircuit model when discussing the importance of the brainstem for consciousness, as the model posits brainstem/forebrain “arousal systems" that control many thalamic and cortical neurons.

Reviewer #2 states: “Which part of the brainstem have included in the atlas? Is it whole brainstem or any specific part (e.g, midbrain). Author could provide detail of chosen brainstem areas/atlas and why they chosen that.” and “Author discussed brainstem as a whole. However brainstem parts are more interesting in the respect of consciousness. Author should address, if there is specific part of brainstem (e.g, midbrain) is associate with conscious process.

We have adjusted a statement in the discussion to further explain the nature of the HCP parcellation, adding the following sentence: “The HCP parcellation, which was built into the preprocessed data that we used, is rather broad and does not parcellate known regions like the brainstem or the thalamus into subregions like the medulla, pons, or thalamic nuclei.”

We have also added the following sentence in the discussion to provide a more detailed explanation of why we chose not to parcellate it beyond what was provided in the HCP preprocessing: “Since our goal was to examine how the brainstem generally fits into the structural connectome, we did not parcellate it further, but a finer parcellation could alter the network even more and may shed light on where the high structural connectivity that we see in the brainstem is concentrated.” 

Reviewer #2 states: “The author used only healthy subject at awake state not in any alter conscious state (e.g., DOC patients), so I think to infer that a region have higher connectivity in healthy subject relay to conscious state/process is not appropriate. Instead of consciousness, it could be any functions.

The Reviewer’s point is taken, and we have accordingly made changes to the text to place less emphasis on interpreting the results in the context of the phenomenon of consciousness. We have also added a paragraph in the discussion conceding that our results do not provide a full picture of the relationship between structural connectivity and consciousness, and that we plan on including DOC patients in future work.

Reviewer #2 states: “Important suggestion: Author should provide the code used and structural connectivity data during or after acceptance of the paper, so some one can replicate the study. Especial the structural connectivity matrix as the author used open data base. This will not only help to reproduce the results also researcher could use structural matrix as a template for functional MRI studies.”

We have explicitly added the github link in the Data Analysis section. This repository contains all our code.

---

## [Decision Letter · Decision Letter 1]

16 Mar 2023

Network analysis of the human structural connectome including the brainstem

PONE-D-22-20864R1

Dear Dr. Salhi,

We’re pleased to inform you that your manuscript has been judged scientifically suitable for publication and will be formally accepted for publication once it meets all outstanding technical requirements.

Kind regards,

Stavros I. Dimitriadis

Academic Editor

PLOS ONE

Additional Editor Comments (optional):

I read carefully your response to reviewer's comment and the revised manuscript, and I

am glad to inform you that your manuscript is provisionally accepted.

I agree with reviewer's recommendation to upload the connectivity matrices to a github repository.

It will make a big impact on your study and future studies on the same direction.

Reviewers' comments:

Reviewer's Responses to Questions

**Comments to the Author**

1. If the authors have adequately addressed your comments raised in a previous round of review and you feel that this manuscript is now acceptable for publication, you may indicate that here to bypass the “Comments to the Author” section, enter your conflict of interest statement in the “Confidential to Editor” section, and submit your "Accept" recommendation.

Reviewer #2: All comments have been addressed

2. Is the manuscript technically sound, and do the data support the conclusions?

Reviewer #2: Yes

3. Has the statistical analysis been performed appropriately and rigorously? 

Reviewer #2: Yes

4. Have the authors made all data underlying the findings in their manuscript fully available?

Reviewer #2: No

5. Is the manuscript presented in an intelligible fashion and written in standard English?

Reviewer #2: Yes

6. Review Comments to the Author

Reviewer #2: The Author have added all the comment's adequately. I do not have further question. Thank you.

I wish to suggestion the authors again, they should provide the structural connectivity matrix data in there Github link, especially as the author used OPEN DATABASE. This will not only help researchers/students to reproduce the results also useful for researchers/students, who are interested to carry out meta-analysis in this domain or looking healthy subjects structural matrix to use as a template.

7. PLOS authors have the option to publish the peer review history of their article (what does this mean?). If published, this will include your full peer review and any attached files.

Reviewer #2: **Yes: **Rajanikant Panda

---

## [Editor Report · Acceptance letter]

29 Mar 2023

PONE-D-22-20864R1 

Network analysis of the human structural connectome including the brainstem 

Dear Dr. Salhi:

I'm pleased to inform you that your manuscript has been deemed suitable for publication in PLOS ONE. Congratulations! Your manuscript is now with our production department. 

Kind regards, 

on behalf of

Dr. Stavros I. Dimitriadis 

Academic Editor

PLOS ONE